# Protein Expression Level Changes in *Weissella koreensis* during Garlic Media Fermentation

**DOI:** 10.3390/biology10060478

**Published:** 2021-05-28

**Authors:** Youn-Jin Park, Myoung-Jun Jang

**Affiliations:** Department of Plant Resources, Kongju National University, Yesan 32439, Korea; coma1052@snu.ac.kr

**Keywords:** *Weissella koreensis* (WK), two-dimensional electrophoresis, qRT-PCR, protein expression, garlic media fermentation

## Abstract

**Simple Summary:**

Garlic is used in cooking and is known to have antibacterial properties due to various organic sulfur components. *Weissella koreensis* is one of the lactic acid bacteria used for fermentation with garlic and is a strain found in kimchi in Korea, one of foods to which garlic is added. In this study, the addition of garlic to the medium did not cause a difference in growth of *W. koreensis*. However, the addition of garlic to the growth medium did produce changes in the amount of specific *W. koreensis proteins*. This study provides basic information about the effect of garlic on fermentation using *W. koreensis*.

**Abstract:**

This study investigated the changes in *Weissella koreensis* (WK) protein expression levels during fermentation in MRS medium supplemented with garlic of WK. WK was first discovered as lactic acid bacteria (LAB) in a Korean fermented cabbage dish known as kimchi. The number of WK cells in MRS medium with garlic (MBCG) and without (MB) after 7 days was 3.55 × 10^10^ and 2.55 × 10^10^ CFU/mL, respectively. To observe the changes in the carbon sources in the media, we measured the glucose, sucrose, lactic acid, and acetic acid levels in each medium (MB and MBCG). Thus, 67.2 ± 2.4 (MB) and 64.2 ± 4.7 (MBCG) mmol^−1^ of glucose were consumed. For sucrose, the level was 3.5 ± 2.2 (MB), and 3.4 ± 2.5 (MBCG) mmol^−1^. There was not much difference in the lactic acid and acetic acid levels at 160.8 ± 0.4 (MB) and 159.2 ± 0.2 (MBCG) and 2.4 ± 0.4 (MB) and 2.2 ± 8.1 (MBCG) mmol^−1^, respectively. After the 7-day fermentation period, two-dimensional electrophoresis (2DE) was used to confirm the protein expression pattern in the WK strain. The results show that the fusA and ssb1 proteins were reduced, and the clpP protein was increased. Afterwards, the expression patterns of the above proteins were confirmed through qRT-PCR. Thus, this study confirms the changes in protein expression levels in *Weissella koreensis* when garlic was added to the media. This study provides basic data for future studies on the major biosynthetic pathways of WK.

## 1. Introduction

A recent investigation on Lactobacillus proteins showed that the 2D electrophoresis technique may be useful in identifying whole proteins that could affect Lactobacillus growth [1]. Bacteria in the genus *Weissella* are Gram-positive, non-sporulating bacteria that belong to lactic acid bacteria (LAB). Their representative forms are short rods, whose length and width are approximately 1.5–2.0 μm and 0.8–1.0 μm, respectively. *Weissella* obtain energy through fermentation, and when grown with sugars such as glucose on substrates, they perform heterolactic fermentation, generate carbon dioxide along with lactic acid, and, depending on the strain, generate acetic acid or alcohol. The new genus *Weissella* was created at the suggestion of Collins et al. in 1993. It consists of six species by combining *Leuconostoc paramesenteroides*, which differs from other species in the genus *Leuconostoc*, and five other species in the genus *Lactobacillus* [2].

New species have been added since then, meaning that, currently, 14 species are registered with the National Center for Biotechnology Information (NCBI) in the United States. Two species were newly reported in 2010 [3,4], and the number will continue to increase because the discovery of new species is expected in the future. Bacteria in the genus *Weissella* have been detected in diverse natural environments including fermented foods such as kimchi and sausages and inside the digestive tracts of humans and animals. *W. soli* has been detected in soil [5], and no pathogenicity has been reported [6]. In 2002, *Weissella koreensis* [7] (WK) was separated from kimchi and reported.

Garlic belongs to the onion genus *Allium* within the lily family *Liliaceae* and contains a large amount of organic sulfur compounds, which intervene in the major bioactive functions of garlic. In particular, the black color of fermented and ripened garlic is a product of the Maillard reaction, a non-enzymatic browning reaction. It contains fructosyl arginine, an antioxidant material [8], and is known to surpass the beneficial effects of raw garlic including the stable lowering of blood pressure, suppressing carcinogenesis in patients with a high blood pressure [9], strengthened immunity [10], and mitigation of physical fatigue [11].

Fermented products both improve the functionality of foods and are ingredients of traditional East Asian medicines; thus, they are expected to be used in the development of products in diverse fields. Chocolates with added fermented and ripened garlic extracts [12] have been reported. Through such fermentation, new materials have been generated, and existing materials could be changed as well. When cultured in a medium containing sucrose, for example, strains within the genus *Weissella* often generate high-molecular weight substances such as dextran. Separated from kimchi, the *W. hellenica* strain generates large glucans with a molecular weight of 203 kDa in an MRS medium containing sucrose, and the optimal production conditions have been confirmed to be pH 5 and 20 °C [13].

This study discovered proteomic variations in the WK strain when WK was cultured together with crushed garlic, which contains diverse organic sulfur compounds besides the polysaccharides above. This garlic-supplemented medium with the WK strain can be used to generate new compounds through WK fermentation in the future.

## 2. Materials and Methods

### 2.1. Garlic Cultivation

The cultivated garlic varieties used in this experiment were cultivated in Daejeong, Jeju, and ‘Namdo’. The average weight of a garlic clove was 5 g, and the planting distances between cultivars were 20 × 10 cm intervals. Other cultivation management was carried out according to the standard garlic cultivation method [14].

### 2.2. Bacterial Strains and Culture Condition

The *Weissella koreensis* (WK) strain was purchased from the Korean Culture Center of Microorganisms (KCTC 3621), and it was cultivated in MRS Broth (Difco Laboratories, sparks, MD, USA) and then stored in 15% glycerol and frozen at −70 °C. The control was MRS Broth (MB), and the experimental cultivation environment was MRS Broth with 10% crushed garlic (MBCG). *Weissella koreensis* (WK) was propagated in MB and MBCG and incubated without agitation at 30 °C for 1 week. After that, cells were harvested by centrifugation at 7000× *g* for 15 min, washed twice, and resuspended in sterile distilled water to give a cell density of 9.32 ± 0.32 log CFU mL. A separate cell suspension was grown for each replicate of each experiment.

### 2.3. Determination of Organic Acids and Sugars

HPLC analyses were carried out for glucose, fructose, organic acids, and ethanol using a Waters System with Empower (Waters Assoc., Milford, USA) [15]. The column was made of YMC pack polyamine (250 nm × 4.6 mm, 4 μm). The mobile phase with a flow rate of 0.7 mL/min was a mixture of water/acetonitrile = 73:27 (*v*/*v*%).

### 2.4. Statistical Analysis

Results were expressed as the means ± standard deviations of at least two independent experiments. To validate the methods, Student’s *t*-test with a *p*-value of < 0.05 was considered significant.

### 2.5. Protein Sample Preparation

After 7 days of growth, the harvested WK samples were suspended in 0.5 mL of 50 mM Tris buffer containing 7 M urea, 2 M thiourea, 4% (*w*/*v*) CHAPS, and 16 μL of a protease inhibitor cocktail (Roche Molecular Biochemicals, Indianapolis, IN, USA) [16]. The lysates were homogenized and centrifuged at 12,000× *g* for 15 min. Fifty units of Benzonase (250 units/μL; Sigma, St. Louis, MO, USA) were added to the mixture and suitably stored at –80 °C until use after quantitation by the Bradford method (Bio-Rad, Hercules, CA, USA).

### 2.6. Two-Dimensional Electrophoresis

For the 2DE analysis, pH 3–10 NL IPG strips (Amersham Biosciences) were rehydrated in the swelling buffer containing 7 M urea, 2 M thiourea, 0.4% (*w*/*v*) DTT, and 4% (*w*/*v*) CHAPS. The protein lysates (500 μg) were cup loaded into the rehydrated IPG strips using a Multiphor II apparatus (Amersham Biosciences) running at 57 kVh. The 2D separation was performed on 12% (*v*/*v*) SDS-polyacrylamide gels.

### 2.7. Coomassie Staining

Following fixation of the gels for 1 h in a solution of 40% (*v*/*v*) methanol containing 5% (*v*/*v*) phosphoric acid, the gels were stained with Colloidal Coomassie Blue G-250 solution for 5 h. The gels were destained in 1% (*v*/*v*) acetic acid for 4 h and then imaged using a GS-710 imaging calibrated densitometer (Bio-Rad, Hercules, CA, USA).

### 2.8. Gel Scanning and Image Analysis

Quantitative analysis of the digitized images was carried out using the PDQuest (version 7.0, Bio-Rad Laboratories, Hercules, CA, USA) software according to the protocols provided by the manufacturer. The quantity of each spot was normalized by the total valid spot intensity. Protein spots were selected for significant variation in expression that deviated more than two folds in their expression level compared with the control or normal sample.

### 2.9. In-Gel Protein Digestion with Trypsin and Extraction of Peptides

The protein spots were excised from the stained gels and cut into pieces. The gel pieces were washed for 1 h at room temperature in 25 mM ammonium bicarbonate buffer, pH 7.8, containing 50% (*v*/*v*) acetonitrile (ACN) [16]. Following the dehydration of the gel pieces in a SpeedVac for 10 min, the gel pieces were rehydrated in 10 μL (20 ng/μL) of sequencing grade trypsin solution (Promega, WI, USA). After incubation in 25 mM ammonium bicarbonate buffer, pH 7.8, at 37 °C overnight, the tryptic peptides were extracted with 5 μL of 0.5% TFA containing 50% (*v*/*v*) ACN for 40 min with mild sonication. The extracted solution was reduced to ca. 1 μL in a vacuum centrifuge.

The excised protein spots from a Coomassie Blue-stained gel were cut into pieces and in-gel digested with trypsin as described previously [17]. The gel pieces, each in a 1.5 mL microtube, were destained with 120 μL of a 1:1 mixture of 30 mM potassium ferricyanide and 100 mM sodium thiosulfate with vigorous shaking. The destained gel pieces were washed three times with deionized water for 15 min with gentle shaking and once with 120 μL of 50% acetonitrile/25 mM ammonium bicarbonate, pH 7.8, for 10 min with shaking. The washed gel pieces were then soaked in 50 μL of acetonitrile for 5 min. After removal of the liquid, the gel pieces were dehydrated in a SpeedVac for 10 min. The dried gel pieces were rehydrated in 10 μL (12 ng/μL) of sequencing-grade trypsin solution (Promega, WI, USA). After a 45-min incubation on ice, the supernatant was discarded and replaced with 10 μL of 20 mM ammonium carbonate. Following overnight digestion at 37 °C, 10 μL of 0.5% (*v*/*v*) trifluoroacetic acid in 50% acetonitrile was added, and the tryptic peptides were extracted by sonication for 40 min with mild energy in an ultrasonic water bath. The extracted solution was reduced to ca. 1 μL in a vacuum centrifuge.

Prior to mass spectrometric analysis, the resulting peptide solution was subjected to a desalting process using a reversed-phase column [18]. A constricted GEloader tip (Eppendorf, Hamburg, Germany) was packed with Poros 20 R2 resin (PerSpective Biosystems, MA, USA). After an equilibration step with 10 μL of 5% (*v*/*v*) formic acid, the peptide solution was loaded onto the column and washed with 10 μL of 5% (*v*/*v*) formic acid. The bound peptides were eluted with 1 μL of α-cyano-4-hydroxycinnamic acid (CHCA) (5 mg/mL in 50% (*v*/*v*) ACN/5% (*v*/*v*) formic acid) and dropped onto a MALDI plate (96 × 2; Applied Biosystems, Forster City, CA, USA).

### 2.10. Identification of Proteins by LC-MS/MS

The resulting tryptic peptides were separated and analyzed using reversed-phase capillary HPLC directly coupled to a Finnigan LCQ ion trap mass spectrometer (LC-MS/MS) [19]. Both a 0.1 × 20 mm trapping and a 0.075 × 130 mm resolving column were packed with Vydac 218MS low-trifluoroacetic acid C18 beads (5 μm in size and 300Å in pore size; Vydac, Hesperia, CA, USA) and placed in line. Next, the peptides were bound to the trapping column for 10 min with 5% (*v*/*v*) aqueous acetonitrile containing 0.1% (*v*/*v*) formic acid, and then the bound peptides were eluted with a 50-min gradient of 5–80% (*v*/*v*) acetonitrile containing 0.1% (*v*/*v*) formic acid at a flow rate of 0.2 μL/min. For tandem mass spectrometry, the full mass scan range mode was *m*/*z* = 450–2000 Da. After determination of the charge states of an ion on the zoom scans, product ion spectra were acquired in the MS/MS mode with a relative collision energy of 55%. The individual spectra from MS/MS were processed using the TurboSEQUEST software (Thermo Quest, San Jose, CA, USA). The generated peak list files were used to query either the MSDB database or NCBI using the MASCOT program (http://www.matrixscience.com, accessed on 30 November 2020). For modifications of methionine and cysteine, a peptide mass tolerance at 2 Da, an MS/MS ion mass tolerance at 0.8 Da, allowance of missed cleavage at 2 Da, and charge states (+1, +2, and +3) were considered. Only significant hits as defined by the MASCOT probability analysis were considered initially.

### 2.11. RNA Isolation

For the identified gene expression profiles, total RNA was extracted from WK strains grown in whole MB and MBCG media. Total RNA extractions were prepared using RibospinTM^II^ Kit (Geneall Biotechnology, Seoul, Korea). The RNA quality was measured by an Agilent 2100 bioanalyzer using RNA 6000 Nanoship (Agilent Technologies, Amstelveen, The Netherland). The RNA quantification was confirmed with an ND-2000 spectrophotometer (Thermo Inc., Madison, WI, USA).

### 2.12. Real-Time PCR Analysis

For the qRT-PCR analysis, RNA (1 μg) isolated from the WK strains grown in the MB and MBCG media was converted into cDNA using a Power cDNA Synthesis Kit (iNtRON Biotechnology, Seoul, Korea). The cDNA synthesis process was started at 42 °C for 60 min, followed by incubation at 95 °C for 5 min to terminate the cDNA synthesis reaction. The 18S (18S ribosomal protein) gene expression of the WK strains served as the housekeeping gene for normalization of qRT-PCR [20]. After dilution of the cDNA, qRT-PCR was performed with Rotor-Gene Q 2plex HRM (Qiagen, Hilden, Germany) using the Rotor-Gene SYBR Green PCR Kit (Qiagen, Hilden, Germany). fusA (5′-TGGCAGAGGAGAAGTACAAC-3′), ssb1 (5′-GCACGTCGTTACACCTTATC-3′), and clpP (5′-ACGTGGAGAGCGTTCATA-3′) were synthesized by Bioneer (Daejeon, Korea). The PCR reaction condition was as follows: denaturation at 95 °C for 10 min, followed by 40 cycles of denaturation at 95 °C for 10 s and annealing at 72 °C for 15 s. The 1-min real-time PCR experiments were conducted with three replications of each gene-specific primer for the 2D electrophoresis selected proteins. The relative gene expression was calculated using the 2^−ΔΔCt^ method compared to the MB sample with the MBCG condition as the control [21].

## 3. Results

### 3.1. Changes in Organic Acids in the MB and MBCG Media during WK Fermentation

Cell growth depends on the chemical changes that take place in MB and MBCG that contain WK after 1 week of fermentation at 30 °C, including the evolution of sugars and organic acids (Table 1). The WK populations in MB and MRCG are presented in Table 1. The cell number increased with the fermentation time and ranged from 10^7^ to 10^10^ (CFU/mL).

In the different media (MB and MBCG), the utilization of sugars and glucose began immediately after the growth, accounting for a major part of the total consumption. Sugars were consumed slower thereafter, and consumption finally stopped before complete depletion at 3 days. No sucrose was degraded. At this time, significant amounts of lactic and acetic acids were produced by the transferred microorganisms. The analytical balance of lactate from the glucose consumed was almost twice higher than the theoretical expected value for the homolactic behavior. Acetic acid was produced, accounting for 40% of the degraded substrates. However, the formed acetic acid accounted for almost all the citric acid consumed, suggesting that the acetic acid derived was exclusively from the citrate.

### 3.2. Protein Expression Patterns of the WK Fermented in the MB or MBCG Medium

We utilized 2DE to obtain data on the changes in the individual protein abundances for WK grown in various media. The seven protein spots per gel were further excised for in-gel digestion by trypsin, and three of them were successfully identified by MALDI-TOF-MS (Figure 1). Three protein spots were differentially expressed by more than ± 1.5-fold according to the PDQuest™ software. Of these three protein spots, one was up-regulated and two were down-regulated in MBCG compared with MB. One protein, ATP-dependent Clp protease proteolytic subunit, was found to be up-regulated, and the two other proteins, Elongation factor G and Phage single-strand DNA binding protein, were down-regulated (Table 2).

Two-dimensional electrophoresis gel stain by Coomassie Blue stain: of the three protein spots, one was up-regulated and two were down-regulated in the MBCG medium growth compared with the normal MB medium.

### 3.3. Validation of the 2D Electrophoresis Data by qRT-PCR

The WK protein expression levels of the three proteins identified in 2DE were analyzed by qRT-PCR. 18S RNA, a housekeeping gene, was used to normalize their expression level (Figure 2).

In the MBCG and MB media, the fusA and ssb1 genes were down-regulated, and the clpP gene was up-regulated in all medium conditions. However, the relative expression level (2^−ΔΔCt^) of fusA was −1.32 in the MB condition and −2.12 in the MBCG condition, and there was a difference of 0.8 or higher between the two, and Ssb1 was found to be 0.61 between the MB and MBCG media conditions. In clpP, the MB and MBCG conditions showed a relative expression level (2^−ΔΔCt^) of 1.83 and 2.41, respectively, and the difference was 0.58 in the expression pattern by growth condition.

## 4. Discussion

In this study, the number of cells during WK growth in the MB and MBCG media was confirmed. Additionally, the consumption of glucose, sucrose, lactic acid, and acetic acid and the growth of WK cells in the two media were verified, and a difference in the protein pattern was also confirmed between the two media. In Lactobacillus, fermentation products containing new functional substances are produced through various metabolites through the fermentation process [22]. These experiments verified the WK proteins of fusA, ssb1, and clpP. Among them, clpP increased by 2.02 times; fusA decreased by 1.5 times; and ssb1 also decreased by 1.39 times.

Elongation factor G (EF-G) is a translational GTPase catalyzing two different steps of protein synthesis [23]. First, EF-G is needed for the translocation of tRNAs and mRNA with respect to the ribosomal 30S subunit to make a new mRNA codon available for decoding [24]. Second, EF-G acts together with ribosome recycling factor (RRF) in splitting the ribosomal post-termination complex. In both steps, GTP hydrolysis by EF-G is used as an energy source, and in both cases, FA prevents the release of EF-G from the ribosome after GTP hydrolysis [25]. The results of this study show that the fusA protein was decreased 1.5-fold when it was processed with the crushed garlic. This result was obtained because the WK strain was grown in the crushed garlic MRS Broth which interfered with the catalytic mechanism of GTPase. Ssb1 is a single-stranded DNA binding protein that is an essential component of the DNA repair machinery in eukaryotes. Recent studies have shown that SSB1 appears to protect newly replicated leading- and lagging-strand DNA of telomeres [26]. Our data show that the SSB1 expression level decreased 1.39 times. The Clp protein with a role in the Clp signaling pathway in lytic enzyme production, antifungal activity, and biocontrol efficiency has been reported in *L. enzymogenes* C3 [27]. Our data suggest that crushed garlic has a biocontrol effect on WK fermentation.

This study reveals by WK proteomic analysis the striking effect of crushed garlic. Microbial cell performance during fermentation processes is a major concern all over the world. Lactic acid bacteria (LAB) with many industrial applications for fermented foods or functional foods, such as kimchi, are usually processed at low temperatures by spontaneous fermentation through a process that enables the growth of various lactic acid bacteria including *Leuconostoc (Lc) mesenteroides, Lc., Lc. citreum, Lc. inhae, Lc. gelidum, Lc. carnosum, Lactobacillus (Lb.) brevis, Lb. curvatus, Lb. plantarum, Lb. sakei, Lactococcus lactis,* and *W. koreensis* [28]. The fermentation process converts odorous, harsh, and irritating compounds present in garlic into stable and safe sulfur compounds, such as S-allylcysteine and S-allyl mercaptocysteine [29]. Fermented garlic powder has been reported to exhibit greater anticancer, immunity-enhancing, cholesterol-reducing, and antioxidative activities compared to intact garlic [30]. The lactic acid bacteria in kimchi contribute to the organoleptic and nutritional properties of this food by means of producing organic acids, bacteriocins, vitamins, and flavor compounds [31]. This results in the desirable health-promoting and sensory characteristics of kimchi [32]. Many LAB strains have the capacity to ferment different carbon sources when cultivated in MRS Broth. This capacity is modified during growth in media that mimic various food ecosystems. Overall, osmotic stress by the high level of carbohydrates inhibits the metabolism of glucose and sucrose and induces alternative metabolic pathways [33]. In conclusion, the crushed garlic influenced the WK protein pattern.

## Figures and Tables

**Figure 1 biology-10-00478-f001:**
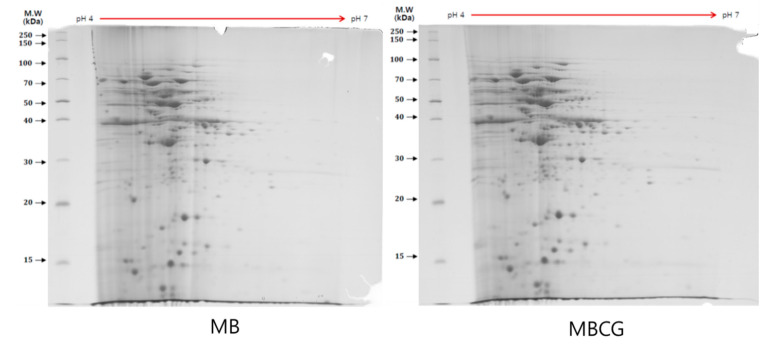
WK protein expression patterns of the MB or MBCG medium.

**Figure 2 biology-10-00478-f002:**
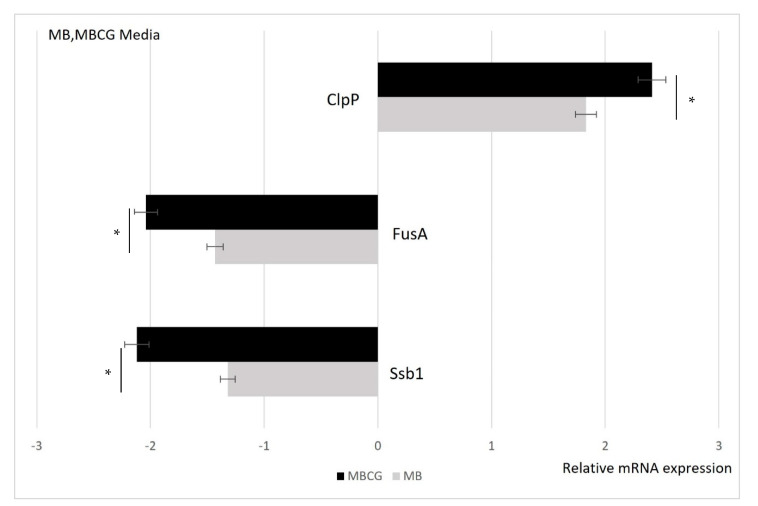
qRT-PCR analysis of the three WK proteins identified in 2DE. Values with an asterisk (*) in the same column are significantly different when compared to those detected in the MB and MBCG media (*p* < 0.05).

**Table 1 biology-10-00478-t001:** Changes in the concentrations of the sugars and organic acids in the MRS Broth (MB) and MRS Broth with crushed garlic (MBCG) during fermentation at 30 °C.

Sample	Fermentation Time (Day)	Cell Number (CFU/mL)	Sugars and Organic Acids (mmol^−1^)			
			Glucose	Sucrose	Lactic Acid	Acetic Acid
MB	Day1	2.23 × 10^7^	98.0 ± 4.2	3.2 ± 2.9	0	0
	Day3	3.22 × 10^8^	70.1 ± 3.6 *	3.3 ± 3.3 *	142.1 ± 1.5 *	1.5 ± 2.3 *
	Day5	4.89 × 10^9^ *	68.1 ± 3.1	3.4 ± 2.4	158.1 ± 1.2	2.2 ± 2.5
	Day7	2.55 × 10^10^	67.2 ± 2.4	3.5 ± 2.2	160.8 ± 0.4	2.4 ± 0.4
MBCG	Day1	1.23 × 10^7^	97.0 ± 3.8	3.1 ± 3.1	0	0
	Day3	2.82 × 10^8^	71.2 ± 2.6 *	3.2 ± 3.8 *	140.1 ± 2.4 *	1.4 ± 1.1 *
	Day5	5.89 × 10^9^ *	66.1 ± 3.3	3.3 ± 3.7	157.2 ± 1.5	2.2 ± 2.4
	Day7	3.55 × 10^10^	64.2 ± 4.7	3.4 ± 2.5	159.2 ± 0.2	2.2 ± 8.1

Values with an asterisk (*) in the same column are significantly different when compared to those detected in the MB and MBCG media (*p* < 0.05).

**Table 2 biology-10-00478-t002:** WK proteins identified in different growth conditions (MB and MBCG).

Spot No.	Uniprot Entry Name	Gene Name	Protein Name	Theoretical pI/Mr	Moscot Score
**1**	F8I065	fusA	Elongation factor G [*Weissella koreensis* KACC 15510]	4.67/77997	219
**2**	J9W094	ssb1	Phage single-strand DNA binding protein [*Lactobacillus buchneri* CD034]	6.93/20293	105
**3**	F8HZ75	clpP	ATP-dependent Clp protease proteolytic subunit [*Weissella koreensis* KACC 15510]	5.52/21979	168

pI = isoelectric point; Mr = molecular weight.

## Data Availability

Not applicable.

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
