# Peer review of "Protein Expression Level Changes in Weissella koreensis during Garlic Media Fermentation"

_biology, 2021, doi:10.3390/biology10060478_

Round 1
Reviewer 1 Report
Biology Review Report
General comments: -
This study is very interesting and has a scientific topic with a great impact on the field. The manuscript will be suitable for publication after taking care of the following minor comments.
Detailed comments:
1-The English language and writing style needs to be adjusted. Please check the manuscript for English mistakes, especially spacing-
2-Line 7 please change galic to garlic
Abstract
1-This section is missing the direct aim of the study. Please state the aim of the study clearly in this section.
2-Please avoid using the personal pronouns (I, We,) as it was found in line 241 (We verified) Please apply this rule throughout the manuscript.
Keywords:
-Please add protein expression, garlic media fermentation, and WK to the keywords
Please change dimentiona to dimensional
Introduction
-The introduction doesn’t provide sufficient background and it is missing enough relevant references
-This section needs to be elongated and enriched with more background about this topic.
Materials and Methods
-Please change Material & Methods to Materials and Methods
Results:
Table 2 please change mmol-1 to mmol-1
please adjust the spacing and table style format
Line 194 and Line 195: - The *in the table footnotes need to be included in the table
Line 207 please add a space after the comma to be MB, MBCG
Discussion:
This section needs to be more organized, rewritten in detail, and fully discussed with related research.
References
This section needs to be UpToDate. Please add research from the last 5 years' citations.
Author Response
Dear reviewer
we changed our manuscript to your comment.
Thanks for your kind comment about our manuscript.
Best regards,
Youn Jin Park

Reviewer 2 Report
Review result of the article ‘Protein expression level changes in Weissella koreensis during 2 garlic media fermentation’ by Park and Jang
This study examined the effects of the medium supplemented with crushed garlic for the growth of Weissella koreensis and its product, lactic acid, production. There is less difference in growth and lactic acid production when grown in garlic supplemented medium. The proteins identified using 2-D electrophoresis and LC/MS/MS analysis are Clp protease (upregulation under MBGC) and the other two ones (downregulation).
Based on the following comments, it cannot be accepted in the present form.
Several concerns need to be clarified.
Sugar consumption and lactic acid production are fast within 1 day and then slow down. It is important information for the subsequent analysis of protein and gene expression. However, the times for protein assay and qPCR analysis are not indicated. More importantly, the reason(s) for why the protein and mRNA PCR are performed at the specific time are is needed.
Although the functions of the three significantly regulated proteins and their gene expression (qPCR) have been discussed, the significant effect of MBGC is not discussed, particularly why these proteins are higher or lower in MBGC medium than MB medium. It is essential for this study.
Other minor comments:
Title: expression
Line 97: Consider the use of Coomassie Blue R-250 due to Coomassie Blue R-250 is better than Coomassie Blue G-250
Line 193: Check the format, mmol-1?
The statistical analysis is essential for the identification of the difference of sugar and organic acid concentrations between MB and MBCG, for example, t-test for the data between MB and MBCG, or overall experimental results using 2-way ANOVA analysis and posthoc test. It provides evidence for ‘not too much variations’ mentioned in Lines 236-237 in the Discussion section.
Line 213: indicate the * for the significant statistical difference in Figure 2 according to the t-test mentioned in Materials and Methods.
Line 222: explain ‘or higher between the two’
Lines 237-238: Because the growth and C compositions in the medium are not significantly different between the two media, it is hard to make sense why the proteome analysis is performed. It needs a solid reason.
Lines 227-235: This is the result that is already described in the Result section. Delete it or modify it.
Author Response
Dear reviewer
'Sugar consumption and lactic acid production are fast within 1 day and then slow down. It is important information for the subsequent analysis of protein and gene expression. However, the times for protein assay and qPCR analysis are not indicated. More importantly, the reason(s) for why the protein and mRNA PCR are performed at the specific time are is needed.'
A : The protein in this study was sampled on day 7 and the total RNA was also extracted atthe same time. It will be added in research and papers on WK protein for lactate sugar consumption and garlic addition over time.
Although the functions of the three significantly regulated proteins and their gene expression (qPCR) have been discussed, the significant effect of MBGC is not discussed, particularly why these proteins are higher or lower in MBGC medium than MB medium. It is essential for this study.
A : It is determined that various substances will have an effect when adding Garlic, so we plan to prove it through experiments using the most stable garlic-derived substances in the future.
And we changed our manuscript for your comments.
Best Regards
Youn JIn Park

Reviewer 3 Report
Reviewer’s Comments:
The manuscript "Protein expresssion level changes in Weissella koreensis during garlic media fermentation" by Park et al presents a case study that evaluated the effect of crushed garlic on Weissella koreensis proteomic analysis.
The following major revisions are required:
- Figure 2 – please label the x- and y-axes.
- ‘Statistical analyses’ section is missing from the methods section. Please mention the statistical tests used in this study.
- Please include the p-values in results section.
- Please mention in the results section if the values presented in Table 2 are Mean ± SEM or Mean ± SD.
- Please include the catalog numbers of primers used in this study. If primers were designed in-house, please include the primer sequences.
- On page 7, line 221, the authors mention that the expression pattern was ‘-1.32’ and ‘-2.12’. Are these delta Ct or Ct values. Please clarify.
- Some of the numerical values presented in the paper lack unit and therefore are difficult to follow. Please make sure you clarify what is being presented.
- Please add a brief paragraph on “future directions to this study” at the end of the discussion/conclusions section.
- Please be consistent with the style of references. For example: page number style in references #24 and #25 is different.
- Please proofread for spelling and grammatical errors.
Author Response
Dear Reviewer
We changed our manuscript every your comments
Thanks for your rewiewing.
Best regards
Youn Jin Park

Round 2
Reviewer 2 Report
Most of the comments have been considered. However, minor comments are needed to be carefully checked again.
- Table 2: Sugars and organic acids (mmol-1) are for glucose~acetic acid, but not for cell number. The format of 2.23 × 10^7 is not commonly used that '7' is superscript, not ^7. Correct them.
- line 215: under MB or MBCG medium. also, line 226
- line 242: (ΔΔCt)? 2ΔΔCt.
- line 253: insert a space between process(22). Check it through the text.
Author Response
Dear reviewer.
First of all, we would like to once again thank the reviewers for all the meticulous editing.
I would like to express my gratitude to the reviewers' meticulous comments for their great role in enhancing the quality of this thesis.
First of all, we corrected all four points and completed marking it easily with a memo.
Please check and reviewing.
Best reguards
Youn Jin Park
